# Analysis of Environmental Issues with an Application of Civil Complaints: The Case of Shiheung City, Republic of Korea

**DOI:** 10.3390/ijerph16061018

**Published:** 2019-03-20

**Authors:** Kyunghun Min, Baysok Jun, Jaehyuck Lee, Hong Kim, Katsunori Furuya

**Affiliations:** 1Department of Environment Science and Landscape Architecture, Graduate School of Horticulture, Chiba University, Chiba 271-8510, Japan; kyunghunm@gmail.com; 2Bureau of Ecological Research, National Institute of Ecology, Seocheon 33657, Korea; baysokjun@nie.re.kr (B.J.); ilandscape@nie.re.kr (J.L.); 3Graduate School of Horticulture, Chiba University, Chiba 271-8510, Japan; hongkim@chiba-u.jp

**Keywords:** civil complaints, citizen information, local issues, issues mapping, keyword analysis

## Abstract

The aim of this study was to better identify the information generated by citizens and to explore the regional social phenomenon whereby higher quality urban services focused on understanding regional issues are promoted. Citizens voluntarily and continuously communicate with local government both online and offline. We wanted to determine how civic information can be applied to urban planning. We selected Shiheung City, Republic of Korea, as our study area, as the city is formed of various types of land use: industrial areas, agricultural land, and residential areas. This area is facing developmental pressure with released development-restricted areas, and has been environmentally damaged by industrial complexes. We conducted a semantic network analysis of the top 10% most commonly used nouns in civil complaints to determine the keywords. Each thematic map we created was based on geographical information to explain the temporary, continuous, and chronic issues. The chronic problems were discussed in relation to the regional development process. The process of identifying and analyzing local issues by analyzing information voluntarily provided by citizens plays an important role in government-led urban management planning and policy formation and can contribute to decision making in the development of future urban policies.

## 1. Introduction

Cities are growing on a global scale, presenting both opportunities and challenges to sustainably improve the quality of life for the residents [1]. As of 2019, almost half of the world’s population resides in urban areas, and this ratio is expected to increase to 68% by 2050. This is called urbanization, which is the increasing trend of the population to migrate toward cities [2]. Urbanization contributes to economic development by promoting economic growth, but causes changes in land cover and land use, and affects crime and disease rates [3,4,5]. Many modern cities have sufficient secure convenience amenities, infrastructure, and green space to fulfill the needs of citizens, but rapidly developed cities may face severe environmental issues in resources-dependent areas [6,7]. Therefore, the merits and drawbacks of urban development may differ depending on regions within cities [1].

The relationship between urbanization and urban services is hard to associate with the interaction with various environments, such as politics, society, and the environment. Deep understanding and management of human–environment interactions is advantageous for identifying the sustainable flow of the interdependent social and ecological balance within a complex city [8,9,10]. A plan for a multidisciplinary approach could improve the interactions of humans and the environment and mitigate the harmful side effects of urbanization [1,11]. Considering regional environmental thinking provides an opportunity for local development without affecting human welfare [12]. The quality of urban life should be considered by understanding the various functions and roles of the environment [13]. Urban policies should be supported by analysis, which contributes to the understanding of citizens.

To improve urban services for citizens, the voluntary participation of many citizens is required to direct the urbanization process and management policy, and governments and organizations should actively encourage participation opportunities. Citizens’ consciousness regarding the trustworthiness of the government significantly impacts many areas of public administration [14,15]. Citizens’ participation has increased both online and offline in local policy development in various urban fields. In offline participation, the communication involves face-to-face and phone call conversations which are securely closed with the security of personal information ensured, whereas citizens are more likely to express their sincere thoughts and feelings anonymously online [16]. The former can help determine the exact opinions of each citizen, whereas the latter enables an understanding of citizens’ perceptions and trends regarding local challenges. Both kinds of communication, which include voluntary citizen participation, should be analyzed to improve urban services.

Many ways can be used to obtain the opinions of citizens: public hearings, polls, citizen/public advisory committees, etc. Although the means vary depending on the purpose, these methods may be tied to the time and place of restriction, and only a few specific representatives may be given the opportunity to speak, which constrains the acquisition of a variety of information [17]. To obtain a wide range of diverse of information, online methods should also be implemented.

Considerable progress in data collection technology has occurred over the past decade worldwide. Data collection has changed from a previously centralized government and surveying offices to a voluntary provision of data by citizens [18]. Cities are rapidly changing, and urban people are producing enormous amounts of data in real time. These data are regarded as a new type of data and are processed as a basis for management planning to address problems faced by municipalities and local organizations. This processing could be used to improve the management of urban services [19]. Major cities around the world are releasing maps and reports that could be shared with citizens. Each state in the United States created a crime map that provides a detailed analysis of the locations and types of the crime based on the state police. The maps are used to help prevent crimes before they occur in these locations, and these maps enable the identification of the crime types in each area. New York City in the United States produces a noise map based on complaints, social media, and other data, revealing the distribution of noise in each area [20,21]. These data types and thematic maps regarding the urban environment contribute to the convenience and safety of urban residents through the promotion of urban services and the prevention of foreseen challenges.

With advances in information and communication technology (ICT) and the spread of smart devices and mobile systems, introducing new applications and services [22], residents can now better participate in urban management and development compared to in the past [23]. Social network services (SNSs) are considered a good example of the latest data and dedicated applications (apps), such as complaint apps. SNSs provide a faster and more convenient process than in the past. SNSs also support accurate geographic information using tools such as smartphone apps, and most tools are accessible and available to anyone for free. The information collected from this communication space can be generated and used as a new data type [24,25]. Given that local data is expressed in a map based on data focused on a region, any resident can easily understand their area’s overall environment, including whether their neighborhood is attractive or problematic. Although the limitation in the mapping process is that the data are obtained from users’ daily lives, analysis of these data is more effective in terms of time and cost than the past given the use of citizens’ opinions in an acquisition of real-time data.

Social networks are actively being studied in various fields, such as education, labor (market), and health [26,27,28,29]. This analytical process plays a key role in a city, as the process provides the understanding as to how use space and how citizens think about space [25]. Local government should actively consider the opinions of citizens when determining how to promote urban services. This trend is also part of the smart city concept, and ICT is being used to improve urban environmental quality for smart city citizens. 

In the Republic of Korea, public discussion sessions, field meetings with citizens, and civil complaints are being used to improve policies and systems. In particular, 36.1% of the results of the civil complaints analysis were reflected in improvements to the policy system by analyzing 2 million civil complaints cases per year [30]. A civil complaint prediction program was implemented to minimize repetitive damage [31]; through regular analysis, the results were used to understand the communications with relevant stakeholders [32]. The government is trying to communicate with citizens by encouraging them to participate in policy creation and change through various channels.

Understanding the social environment issues based on reported citizen information about discontent with urban services and local challenges are identified using the spatial distribution of issues. In particular, we attempt to highlight a new feasible urban policy based on civil information in each local government in the Republic of Korea and encourage citizens to participate more by contributing civil information to help characterize the actual issues in a region. We aimed to determine how different civic information can be applied to urban planning. We selected Shiheung City, Republic of Korea, as our study area, as it contains various forms of land use: industrial, agricultural, and residential areas.

## 2. Materials and Methods 

### 2.1. Study Area

Shiheung is located to the midwest of Gyeonggi province, bordering the Bucheon, Ansan, Hwasung and Namdong districts of Incheon Metropolitan city, and was planned to distribute the population and industry in the metropolitan area. Restricted development areas enacted by national law are the largest in the bordering districts, with most of this area being agricultural land and greenhouses, although the development-restricted area is decreasing (from 111.53 km^2^ in 2014 to 85.846 km^2^ in 2016) and the areas released for development are the highest within the province due to a policy enacted in 2015 (Figure 1). With the implementation of a new urban plan, new towns are being built. The total study area is 138,562 km^2^ and has a population of 439,685 in 176,588 households. As one of the largest national complexes adjacent to the Yellow Sea in the country, their facilities area connected to the neighboring Ansan city in the southern area. Urbanized areas in the city are located near the industrial area in the southern area and the northbound area. Considerable amounts of green land remain within this city, especially when compared to other Korean cities. Of the total area of the city, 63% remains natural environment and green spaces are being created in cities under development. Forest land, paddy fields, and streams comprise a large proportion of this area. Large paddy lands were reclaimed in the 18th century and the inland mudflats, used for salt farming in Gyeonggi province and a national protected area, are connected to neighboring ecological parks. Focusing on paddy lands and mudflats, the National Natural Trust project worked with a local government initiative to improve ecosystem services within the city (Figure 2). According to the ecosystem services assessment conducted by the National Institute of Ecology in the Republic of Korea, the city has high-quality cultural ecosystem services [33].

Even though many ecological resources are evident, environmental ideas have not been considered since the 1980s due to development-oriented policies. National industrial complexes were built on reclaimed land via tide embankment construction, which impacted to marine and terrestrial ecosystems by destroying and damaging wildlife habitats. Without seawater circulation, the surrounding areas were affected by the excessive flow of contaminated wastewater from nearby industrial complexes. In the end, the emphasis on the development of urban policy provided poor services to the surrounding ecosystem.

Until recently, residential areas have been affected by industrial complexes and the construction of the new town in the released development-restricted area. Since 1992, the local government has built buffer green areas 4 km long and an area of 0.692 km^2^, extended by 0.03 km^2^ in 2016, between industrial complexes and residential areas to prevent the spread of air pollutants, the buffers have not had a significant effect of preventing contaminated materials from industrial areas. Other residential areas have been damaged by the construction of large-scale settlements. This has led to growing complaints from citizens related to these developments.

In this study, we considered the opinions and dissatisfaction of citizens on living conditions on the promotion of urban services. Opinions are mostly citizens’ environmental complaints in Shiheung. We formed clusters based on words using Netminer 4 (social network analysis software, CYRAM, Seongnam city, Republic of Korea) and created a visualizing map as occurrence location. The purpose was to express each district’s environmental demand. If there is no place mentioned, it was excluded from the analysis process. The locations mentioned are marked with small dots, but they are easy to understand because each type is displayed locally.

### 2.2. Date Collection and Methods

The data in this study included environmental civil complaints for 3 years (2014–2016), targeting Shiheung city. The data included date and time, occurrence location, and the unsatisfactory situation perceived by citizens. Since complaints were written in consideration of the reporter’s perspective, prejudice or bias could exist, and thus the recorded complaints were collected and used in various ways. We analyzed a total of 1453 records; records with missing or mismatched geographic information were excluded (187 in 2014, 362 in 2015, and 324 in 2016) (Table 1). The data were categorized as 4 types based on reporting channel: phone calls, night-time phone calls, internet, and use of the smart phone app. Phone calls are used for the caller to directly report their complaint to government officials, whereas night-time is off-hours reporting. An opinion on the internet is reported via the government website signboard or upper level authorities, whereas opinions are directly reported using smartphone applications developed by each of the government authorities.

Our analysis was divided into two parts: extracting the considered keywords from the data, and creating a demand map of the keywords. We created a network map for each year based on the selected words using a cleaning process with the top 10% of all nouns in the complaints using by R studio (statistics software, Boston, MA, USA) named KoNLP (Korean Natural Language Processing, R software packaged, Seoul, Korea) to exclude ambiguous and meaningless words. To understand the relationship between keywords, we performed network analysis to determine the connectivity of nodes and links between keywords. The keywords with high connectivity that were located in the core of the network were recognized as important issues through degree centrality analysis. Keywords that acted as bridges between keywords within the network were determined via betweenness centrality analysis, which affects the overall flow. Among them, meaningful keywords with geographical information were employed to create an issue map and determine the areas in which these words are concentrated. In doing so, Netminer 4 software and ArcGIS 10.5 (geographical information system, ERSI, Redlands, CA, USA) were used in the process (Figure 3). 

Each complaint has merits and drawbacks depending on the reporting method: telephone calls immediately connect government servants with the caller and the report’s exact intention can be easily understood, but records depend on government officials. The internet responses do not have a typical format because the reporter writes the content on their own, but accurate descriptions of the occurrence location and exact content are provided. The smartphone application automatically records information of the occurrence location and is efficient, but not free from the set format of the application.

### 2.3. Civil Complaints Management in the Republic of Korea

A management system of complaints in Republic of Korea was launched to increase transparency about public officials’ inadequate behaviors and the financial conditions. This system, known as the OPEN (Online Procedures ENhancement for Civil Applications) system, was introduced by the Seoul metropolitan government in 1999, released to public, and recognized as an internationally efficient system for citizens by the United Nations (UN) [34,35]. Since then, the process of controlling and managing local government has become more transparent through the active participation of citizens. Each municipality and ministry department has developed a civil service system on the Internet and developed web services and smartphone applications to be used by citizens. The opportunities for citizens to participate in policies and planning has increased dramatically compared to the past due the simplicity and speed of feedback.

The Republic of Korea has a complaint reporting system, and each ministry, department, and local government has developed an Internet bulletin, smartphone apps, and a call center to meet citizens’ demands. For the offline reporting system, any citizen who has complaints can report directly to city hall or the provincial office to meet the civil service officer or call that person. Online complaints can be reported via web bulletin boards or smartphone apps with attached video or pictures. Dedicated apps include e-people (*Gugminsinmungo* in Korean) developed by the Anti-corruption and Civil Rights Commission and Life inconvenience report, introduced by the Ministry of Public Administration Security, which can be used more conveniently by automatically providing a location. Citizens can provide a report in many ways and anywhere, and the reported complaints are transferred to the local government or the governing department for review and response. Not all local governments and ministry departments have their own apps, but they have focused on the efficient and fair handling of complaints. This condition has drawbacks for the analysis of the concept and functions of civil complaints, but all processes are developing citizens’ policy participation [36].

## 3. Results

### 3.1. Relationship Analysis between Words in 2014

We analyzed environmental complaints over three years by conducting networking analysis. Significant keywords were identified based on the node and links via degree and betweenness centrality analysis. In 2014, the frequency of the words in decreasing order was as follows: noise (344), stink (241), construction (141), dawn (106), dust (95), and nighttime (91). These words, except for dust, had high values of degree and betweenness centrality. They were recognized as an issue because other words were linked to them and these words were located in the center of the network map. The network map consists of different sizes of nodes and different length links. The node’s size varies with the importance of word: the larger the size, the more important the word. The links’ length denotes the association between each word: the shorter the distance, the more relevant the association (Table 2).

The network map could be categorized into two parts based on noise and stink words. Noise was directly interlinked to construction, dawn, night-time, and holiday, and stink was connected to factory, window, daily life, and livestock, and especially dawn and night-time words associated as a mediator. Considering the overall network flow, noise was recognized as a keyword for the main issue (Figure 4). We predicted noise was related to construction at night-time and dawn and everyday life, such as sleep and music.

### 3.2. Relationship Analysis between Words in 2015

In 2015, in terms of frequency, stink words were the most frequent (451), followed by noise (241), dawn (152), and dust (115) words (Table 3). The stink words with the highest centrality value were directly connected to the words dawn, contaminant, incineration, gas, and fuel; and the contaminant words were interlinked with incineration, fuel, gas, pain, pollution, waste water, and maleficence words. The overall workflow was described with stink in the center of the network map, as shown in Figure 5.

Comparing the network map and the result of betweenness centrality, noise was strongly associated with construction and other words were interlinked to stink. Incineration, which was located stink and contaminant, had a connection with fuel and add facility. Considering the overall flow of the network map, stink was a keyword representing a main issue. Stink was associated with contaminant, gas, and incineration, and fuel, incineration, and add facility were also associated.

### 3.3. Relationship Analysis between Words in 2016

In 2016, the highest-frequency words in the complaints were noise (728), followed by stink (594), construction (255), night-time (173), and dawn (129) (Table 4). Most words were connected to the noise, which was directly interlinked to construction, dawn, nighttime, factory, sleep, residents, and window. Stink also was connected: dawn, factory, sleep, residents, and window, which were located between noise and stink. The overall flow was described, with noise in the center of the network map. Considering the overall network structure, the main issue keyword was noise. The words that had high betweenness degree that were interconnected with noise and stink were shown to play an important role in issue (Figure 6).

### 3.4. Sintering of Words Analysis for Three Years

We analyzed three years of data to determine the keywords that appeared every year. The keywords that were continually extracted were as follows: noise, smell, construction, nighttime, dawn, dust, factory, sleep, window, and pain. The frequency of most of these words was high in 2016 and they all increased after 2014. Noise and smell accounted for a fairly high proportion of the keywords; construction, dust, and factory were the causes; and night-time and dawn provide understanding of occurrence time; the effect on citizens was sleep and pain; and the window seemed to be the mediator (Figure 7).

Although words with high frequency and centrality values were important, analyzing the other words around the keywords was meaningful to help us understand various words related to the keyword. Words excluding the main keywords were as follows: holiday, daily life, site, music, and residents in 2014; add facility, contaminant, incineration fuel, and weather in 2015; and residents, holiday, inconvenience, residence, and daily life in 2016. Overall, the number of keywords was increasing as of 2016. Residents, holiday, daily life, site, and music were extracted in 2014 and 2016, which appear to be related to noise, which was the main keyword for two of the studied years. Chemicals and gas appeared in two years, which also appear to be linked to smell as the main keyword in 2015 and noise in 2016 (Table 5).

## 4. Discussion

The main keywords based on the three years of data from the voluntary participation of citizens represent the city’s challenges. The keywords were used for more in-depth reviews of the location of complaint and the time of occurrence. Every month and year should be considered by city managers to contribute to the monitoring and problem solving in each location and to provide better urban services to the citizens. We visualized the thematic maps applied to the geographic information of each keyword to understand the density of reporting in the study area. Since most complaints were reported as problems occurring within the city, comparison with population distribution was effective for analyzing the concentration of the frequency of complaints. The distribution of population did not change significantly over the three-year period, with an average population of 400,000 inhabitants. Combining the thematic map and the map of the population distribution, the association with the reported location could be identified and, when also considering the network map, the contents of the complaints could be interpreted more clearly.

### 4.1. Local Keyword Issue

#### 4.1.1. Noise Keyword Issue

Noise is one of the most common environmental problems, and its occurrence frequently occurs within the city [37]. Noise is a temporary or permanent harmful sound that affects human hearing and causes discomfort [38], and might pose threats to human mental state. Because it threatens human life constantly, noise must be managed comprehensively.

We identified that noise was the key challenge in 2014 and 2016. Although noise was not a keyword in 2015, it was necessary to analyze the data for all three years since frequency and centrality play important roles in issue analysis. Overall, the reported complaints were mostly concentrated in the summer season, from June to September, and the proportion of noise complaints was the highest in 2016, and decreased slightly in 2015, but increased sharply in 2016 (Table 6).

Comparing the noise map with population density, the most reported complaints came from urbanized residential areas, where much of the noise was concurrent with population distribution. Considering the overall the noise map for the three years, most noise reporting occurred near residential areas. The northern area reported the most complaints in 2014, which then decreased in the next two years, but new noise complaints in the eastern region were reported, and those in the southern region decreased (Figure 8).

As described in Section 3, noise was identified as having a strong connection to construction, dawn, night-time, holiday, and music. Many complaints were reported during the summer season such as, “Please take action because remodeling work in front of my house is underway on a holiday”, “The store’s music is so loud that we cannot sleep at night-time” and “The building near my home has been under construction since early dawn on the holiday, please settle it quickly”. As a result, the noise complaints in 2014 mainly reflected the impact on living. In 2015 and 2016, complaints in new areas emerged. These were perceived as the development of the eastern region was beginning with the construction of the new town at the end of 2014. The noise complaints in 2015 included, “Noise interfered with sleeping due to the construction of an apartment building that began at dawn”, “Adults could not breathe because of the dust from the apartment construction and vehicles, but children are more worried”, and “The noise of the machinery in the next door factory is so loud”. Most of the 2015 noise complaints were due to the large apartment construction sites and industrial complexes. In 2016, the noise complaints were more concentrated in the eastern region. There were many complaints in 2016, such as, “We could not sleep because of the hymns that come from the church every Sunday at dawn”, “It is inconvenient to open the window because of the constant dust from large scale construction”, and “The noise of metal falling from the construction site at night-time is painful”.

Through analysis of the occurrence location and content of the complaints, the noise could be divided into temporary noise, such as a living noise, and continuous noise, such as development and construction noise. Temporary noise can be addressed immediately and quickly, but complaints of continuous noise might be further exacerbated by ignoring citizens’ opinions on urban policy. Through the complaints’ content, we identified that citizens demand systems and facilities to reduce discomfort and pain and demand the halting of construction. The complaints that windows and doors cannot be opened in the summer season support the reason why complaints were concentrated in summer from June to September.

When noise is reported, the government official measures noise on the reported spot. In the case of living noise, more effective measurement methods or safe solutions are required because the complainants may not want to report because of the risk of personal information being exposed. In the case of construction noise, more thorough management and prevention required, such as the installation of a sound barrier and compliance with regulated construction hours. Other cities in Korea have noise-free days or monitoring on site as a measure to reduce noise [39]. Noise distribution maps can help identify noise-sensitive areas and reduce additional costs and time delays due to noise measurements.

#### 4.1.2. Stink Keyword Issue

As a result of increased industrialization and awareness of people’s demands for a clean environment, there is increasing interest in stink. Odours have a profound effect on humans because they are a feature of harmful or toxic substances in the air [40], which have adverse effects on quality of life and entire communities [41]. There are various artificial causes of odours, but thorough regulation and control are required before the damage increases to citizens.

Stink was the main keyword in 2015 and continued to appear as meaningful word along with noise. The reports of stink increased every year, with the highest proportion in August, but with different patterns in each month and year. February 2015 had the highest number of complaints due to smell in the three years, far higher than 2014 and 2016 in the same month (Table 7).

As described in Section 3, smell was strongly association with factory, contaminant, gas, incineration, fuel, chemical, and livestock. Viewed in terms of spatial distribution in 2014, smell complaints were distributed near industrial complexes in the southern regions, and residential areas in the northern region. The reported complaints stated, “I cannot open the window because the nearby factory smells like burning rubber at nighttime”, “The livestock around agricultural land smells too bad. When windy or raining, the smell is worse”, and “It smells at dawn, but I think it is coming from near the factory, please check”. Bad smells were reported in residential areas in 2014. However, the reported locations in 2015 were unusual. Although there were many reported locations in 2015, most of the reported complaints were concentrated in one location. The reports from this location included, “I cannot take a rest with my family in the park because of the smoke and stink from the nearby paper factory”, “I am against the idea that the paper factory will be expanded. Even now, the incinerator facility smells like chemicals, and the expansion does not make sense”, and “I have no idea that what fuel they burn in incinerators and what is the identity of the gas, but it stinks every dawn”. The complaints in February were mostly about the paper factory. In 2016, there were many complaints related to this factory and reports from dwellings compared with 2015. Complaints in 2016 included, “There is constant gas coming out the factory that smells like chemicals”, “The smell of the factory is damaging to all the residents of the apartment. We would like to find out why it smells every night or at dawn”, and “I closed the window because of the chemical smell, but it still smells. I cannot sleep at all”.

Unlike distribution of the noise complaints, smell complaints were concentrated in one location with particular words. The location of the main cause of smell generation was predicted by a particular factory. Although the factory was approved by upper government including establishing stink-reducing measures, the smell is still bothering residents because the plan has not been implemented. The factory owners, citizens, and local government are still negotiating, but no resolution had been reached at the time of publishing this report (Figure 9).

### 4.2. Reporting Channels and Continous Local Issues

As described in Section 2, there are four available reporting channels: daytime phone calls, night-time phone calls, Internet bulletin signboards, and smartphone apps. The overall reporting frequency increased, but each channel has pros and cons. The day and night phone calls, as offline channels, are convenient for reporting complaints immediately because reporters talk directly with the person in charge, so the reporters can realistically convey their feelings and the situation causing dissatisfaction at the time. However, the subjectivity of the person in charge might affect the content of the recorded complaint, and it is difficult for the complainants to check the process or the result in real-time. In addition, if the person in charge does not understand the exact occurrence location being reported, or if the reporter does not describe its location correctly, the process of solving the problem could be difficult and time-consuming. Reporting via Internet bulletin and apps on smartphones, as online channels reported on the Internet, are not tied to place or time. Photos and videos can be attached to provide evidence of the issue, which can be released to other citizens through bulletin board homepages and apps. Although online methods enable convenient monitoring of the process, immediately provide results, and the location of problem can be automatically transmitted from apps or described in detail by the reporter, these methods can be abused or misused by a particular person or a specific problem. The person in charge may struggle to interpret or understand complaints because of the free form nature of the process. Online channels are useful for real-time analysis and mapping compared to offline channels because information is more open and reporting is independent of time and place. In particular, a more well-organized online platform will help citizens report complaints and gather information, enabling managers to easily obtain and analyze data.

We noted that nighttime phone calls had the highest reporting rates and that the words reported then were associated with a particular time: night-time and dawn. Night-time phone calls mean that a complaint was filed by telephone outside regular business hours—a complaint reported at nighttime or dawn. Night-time and dawn frequently appeared around the keywords for all three years, and were mentioned often in the content of actual complaints. It could be inferred that many problems occur at these times, so accordingly, many complaints were reported. This hypothesis is supported by the fact that the number of reports at nighttime is much higher than during the day. In this respect, based on civil complaints, some factories or companies in industrial complexes may be illegally discharging waste or gas, which was the cause of the smell at night-time or dawn. In the past 20 years, stink complaints have intensified every year, and these complaints continue to increase. In 2018, local governments launched a civil environmental watchdog initiative based on citizens to ensure monitoring and crackdown. Further scientific analysis is required as the initiatives included normal non-expert with subjectively assessing the stink. 

The large-scale residential area behind the industrial complex in the southern region poses a social problem. Because this residential area is only 200 m away from the industrial complex, it is easily exposed to noise and smell, as frequently reported in civil complaints. After the construction of the complex as part of the industrial city plan, this residential area was built for workers, but the construction of residential areas near the complex was initially deemed inappropriate.

Chronic problems such as noise and smell are difficult to respond to immediately. The speed with which complaints are addressed varies depending on reporting channels, but the transfer process takes time if the complaint is filed elsewhere than with the local government. Each local government attempts to speed up the processing of civil complaints by granting incentives to officials or notifying officials a few days before the expiration of the processing period limitation. The local governments should regularly analyze the handling status of local civil complaints and predict future civil problems. This process contributes to solving the fundamental cause of the complaint, and is not temporary problem solving that is likely to cause the recurrence of complaints.

### 4.3. Role of Government in Civil Complaints

Systemic management of civil complaints is required to improve urban services. Local governments should deploy a platform or application that can be used to immediately identify all complaints and build complaint maps to prepare for various problems and new complaints via continuous monitoring. Local governments should introduce civil initiatives to engage the citizens in activities by providing expert education. The initiatives should provide data through field surveys conducted by experts that can be used for analysis. Policy makers should organize a framework to apply civil affairs analysis to urban management plans, and regular hearings should be held to determine the opinions and concerns of the residents about the policy. In particular, for chronic complaints and severe problems, the number of government employees needs to be increased, and a systemic process consisting of officials, experts, and initiatives should be implemented to ensure the complaints and problems are addressed. This information gained through these processes will be used as a basis for making decisions on urban development and management in the future by realistically identifying problems that occur within the city. The opinions of citizens will play a key role in improving urban services, opposed to a government-led policy, as citizens enable voluntarily and realistically to raise issues.

The national human rights commission of the Republic of Korea the status of and trends in filed civil complaints in real time by province units, institutions, sex, and age on a website on 30 January 2019. The website also discloses the status of the various classifications being handle, such as living conditions, civil and criminal types, environment, and industry. Although this system is effective for managing and predicting the entire national territory on a government level using keywords, it is difficult for citizens to understand the issues in detail at the city unit level. In this regard, the complaints from city units are required by the local government, especially the chronic complaints that should be handled immediately that require cooperation at the national level.

### 4.4. Comparision with Another Civil Complaints Study

So far, not many studies on civil complaints have been published. We discussed process improvements and the applicability of civil complaints by comparing our study with a recently published paper about civil complaints.

Hong et al. [42] studied crowdsourced big data based on noise complaints from the 311 system, which is the full-time non-emergency telephone services and online platform, in Vancouver, British Columbia, Canada. The utilized 311 data were classified as either “noise complaint case” or “general noise inquiry case” from 2011 to 2016, and were compared with the location and geographic information of noise. More detailed analyses of the possibilities of bias in the data, such as superuser, confirmed that the data was not affected by a small number of specific variables. They found that noise is associated with constriction activities, which are part of urban development. In particular, construction activities hinder the sleep of residents, reflected in the large number of after-hours complaints compared to those during regular hours. The results of the mapping were strongly related to construction and noise in similar locations. Finally, the authors suggested that crowdsourcing based on civil complaints in an open government framework with smart cities can contribute to local and national interests.

Firstly, considering the similarity between the two studies, both identified the value of civil complaints. The data are based on information generated by citizens’ voluntary participation, such as online-based crowdsourcing and big data, and offline data gathered by telephone and direct visits. Secondly, visualized the analyzed data on maps is to identify realistic locations of civil complaints. These annual maps provide an understanding of where occur to these complaints throughout the city. Thirdly, both studies confirmed that civil complaints are strongly related to night-time (night-time and dawn in our case, after-hours in the other study case). Finally, this process suggested that the civil complaints analysis contributes to government and urban policy in smart cities.

There are differences in the studies as well. The approximate work flow is similar in both studies, but the methods are different. In our study, data were analyzed and verified using a high centrality of keywords analysis, whereas the other case applied covariate analysis on the basis of demographics. In addition, in the process of mapping, we used the geographical information of each keyword; however, the other study completed a comparison by reflecting the locations of the complaint occurrences and the construction locations.

In conclusion, our study facilitates understanding the issues experienced by citizens throughout the entire city area, whereas the other study is more useful for concentrating on one issue and analyzing it closely. It is expected that our study process will realistically and clearly help with future applications to statistics or demographics.

## 5. Conclusions

As cities grow, various challenges develop. Good urban development planning contributes to an improvement in the city image by providing higher-quality urban services to citizens; poor urban management policy increases complaints by citizens and environmental problems resulting from development. Urban planning requires active participation of the citizens, and city managers should listen to their opinions to identify any problems.

Advances in technology have enabled citizens to participate in policy development and change and allowing participation to occur faster and more conveniently. Citizens can express their dissatisfaction through various means, both online and offline, to communicate ideas to local governments about urban development and management. A systemic analysis plays important roles in urban policy as citizens’ complaints are continuously generated every day. 

The data used in the analysis are based on three years of civil complaints received both offline and online. We identified the keywords in the complaints for each year using semantic network analysis to extract nouns from the data and analyzed the words associated with the issues. Based on the geographical information of the keywords, thematic maps were established to understand the spatial distribution of the issues. By understanding the areas where the issues were concentrated, we identified temporary and continuous issues. The particular time and location of the chronic problems were discovered.

Noise and stink were identified as the main keywords for the social issues based on the distribution map of the issues. Most of the complaints were filed in and around residential areas. In 2014, noise was associated with living noise and construction, and in 2016, it was strongly related to a large-scale construction area. In 2015, stink was identified that was associated with an industrial complex, and particularly in one area. At that time, the plan for the extension of the factory within the area ceased for a while, but the dispute has not yet been resolved.

Viewed from each network and distribution map, the noise that affects those living in the northern part of the city decreased, the construction noise increased in the eastern part, and smells continued to cause complaints in the southern part. In addition, we confirmed that stink cases at night-time and dawn were associated with stink cases in residential areas around industrial complexes. This result means more crackdowns and thorough management are required on these industries, as it might be illegal for factories to operate after civil servants or managers have left the office, or after certain hours. We confirmed the civil complaints and the high rate of reporting at nighttime and dawn, but this requires more scrutiny.

In the Republic of Korea, the reporting of opinions by citizens, including civil complaints, may be used as a valuable resource for promoting city services and updating or creating policies. These reports are more realistic because the problems directly experienced by citizens are voluntarily filed. In the fields of crowdsourcing and citizen sciences, the information generated by citizens is already recognized as data and used for various purposes. These processes provide opportunities for citizens to engage in policy-related activities.

This analysis process can be applied to a variety of fields, such as crime, fire, and hygiene in each city, and is useful for prioritizing the tasks to be addressed. The establishment and integration of each thematic map are required for the improvement of urban services and management from a long-term perspective, providing the ability to identify or predict complex issues.

We suggest that the analysis of civil complaints can help improve city management. Establishing thematic maps monthly or annually can contribute to the creation of new urban policies. Urban policies addressed by including communication with citizens will be more successful than government-led polices.

The limitations of our study are that, firstly, it does not contain data that have been scientifically investigated, such as noise and smell levels, in the analysis of the issues. That is, the subjective experiences of problems are different based on the individual. Secondly, analysis based on occurrence time is also insufficient because complaints filed by Internet bulletin signboards and apps on smartphones do not verify a reported time, and occurrence time might be inconsistent with reported time. For example, a problem that occurred at night-time might be reported the next day. Finally, not all citizens file complaints in the city, nor are the complaints representative of the complaints of all citizens. Nevertheless, the number of civil complaints is continuously increasing, and citizens demand that their complaints be addressed and resolved. These complaints are not representative of all the issues that occur within the city, but civil complaints might be considered typical samples that can be used to improve the urban services process or planning for citizens.

We propose two methods of mapping in future research. The first involves applying the happiness score of words, such as “Headnometer” [43], to the civil service analysis process. By applying the happiness of each keyword, we could conduct thorough analysis and establish various maps, and understand the local happiness and dissatisfaction through maps. The second method involves using a public participation geographic information system (PPGIS) and comparing the results with the maps of keyword analysis. The result of this method can be great because this is mapping reflects the spatial cognition of the participants based on their experiences. The results reflect general but critical issues of the environment perceived by citizens.

## Figures and Tables

**Figure 1 ijerph-16-01018-f001:**
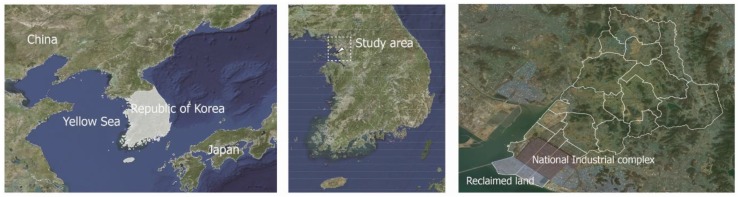
The location of the study area in the Republic of Korea.

**Figure 2 ijerph-16-01018-f002:**
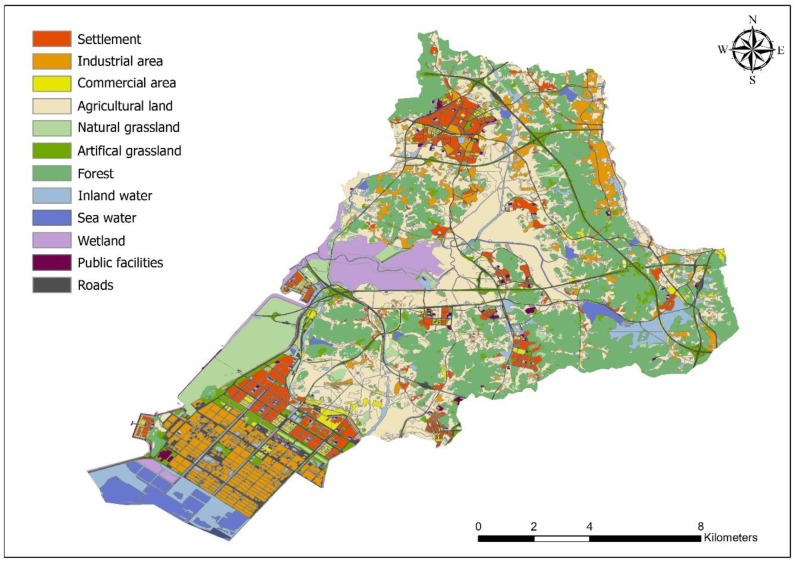
The land cover map of the study area.

**Figure 3 ijerph-16-01018-f003:**
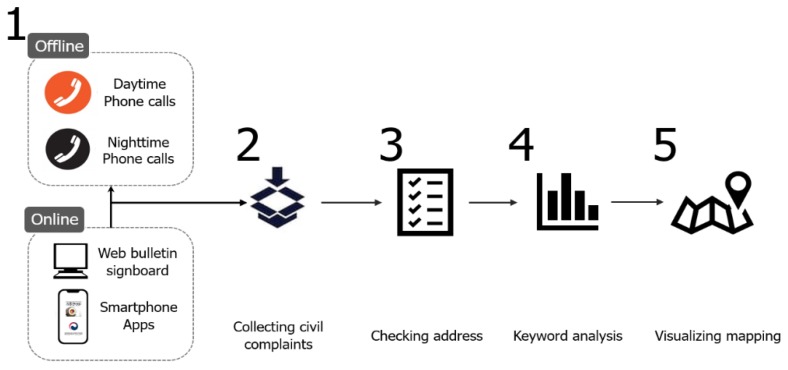
Flowchart of our research.

**Figure 4 ijerph-16-01018-f004:**
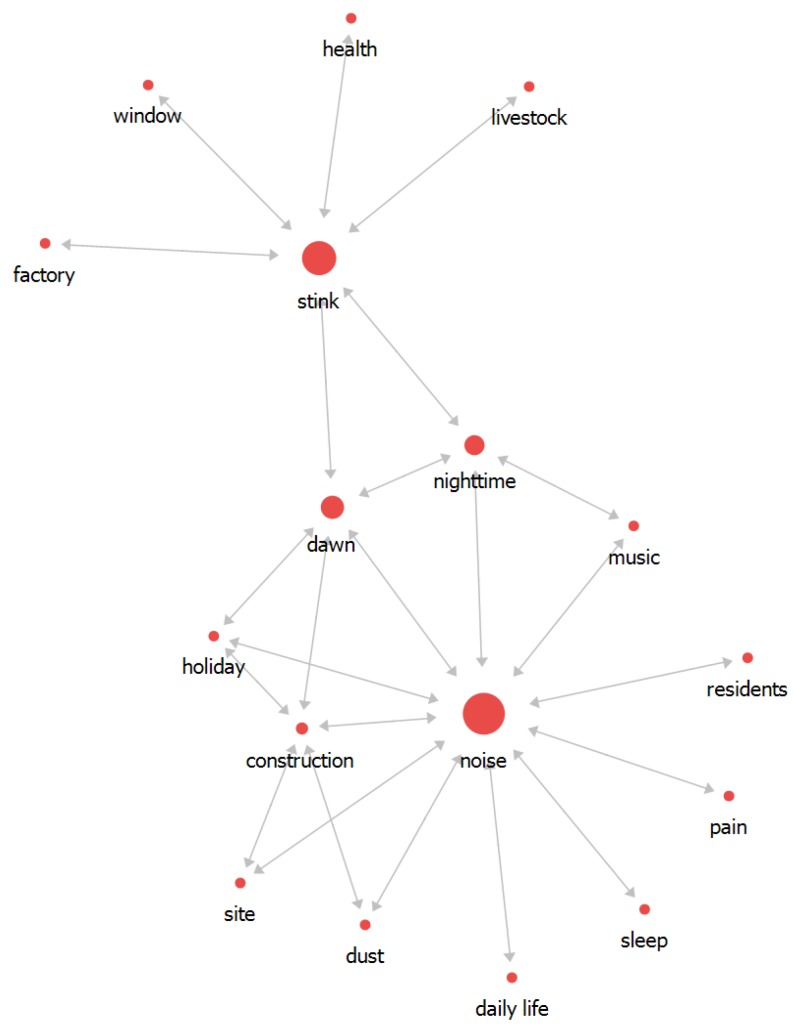
The relationship between words in 2014.

**Figure 5 ijerph-16-01018-f005:**
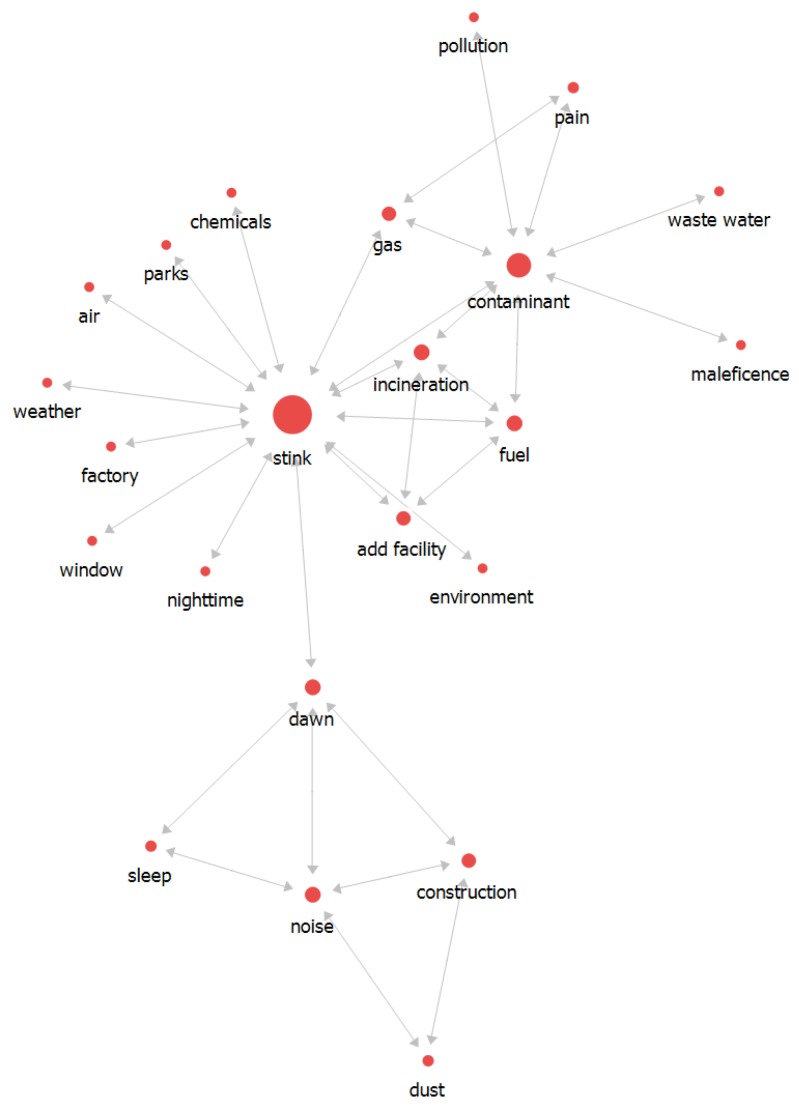
The relationship between words in 2015.

**Figure 6 ijerph-16-01018-f006:**
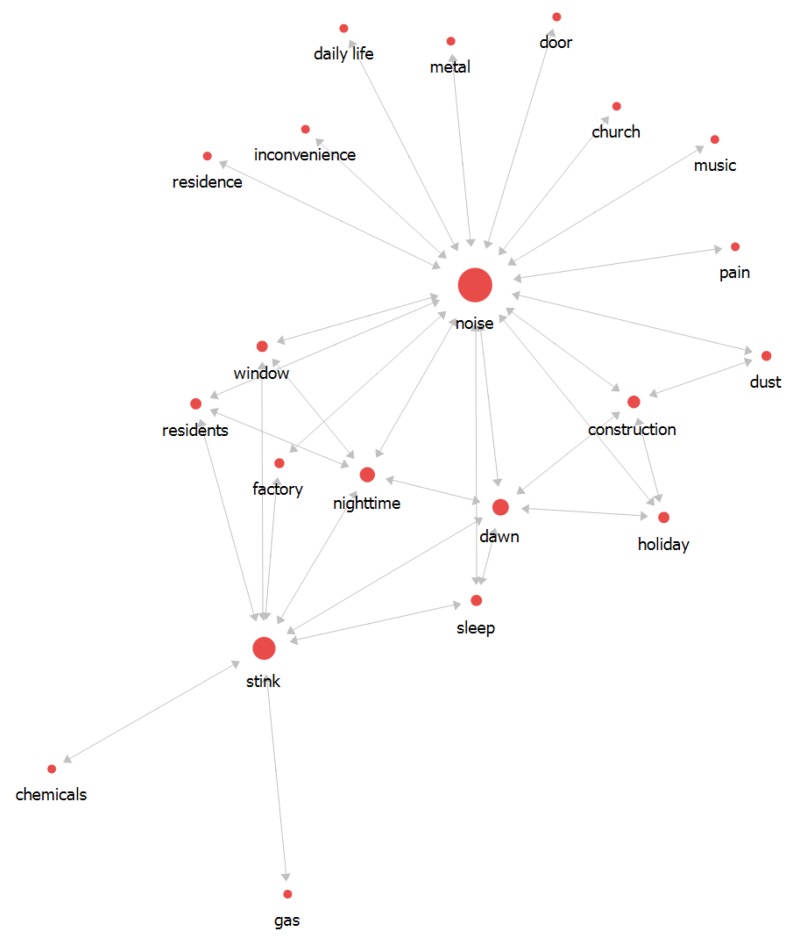
The relationship between words in 2016.

**Figure 7 ijerph-16-01018-f007:**
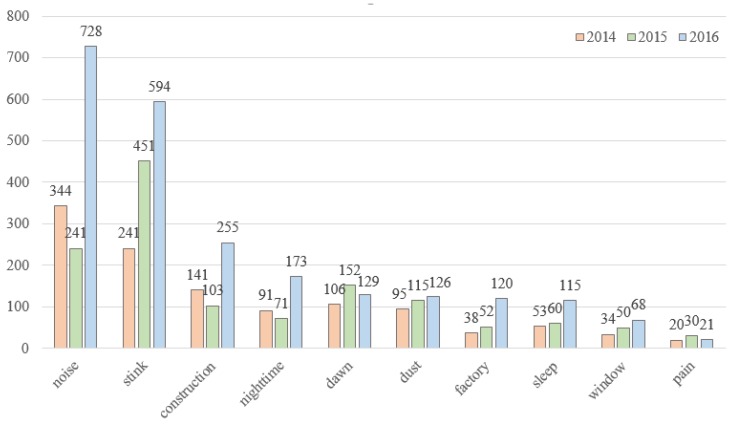
The frequency of words that have been appearing for three years.

**Figure 8 ijerph-16-01018-f008:**
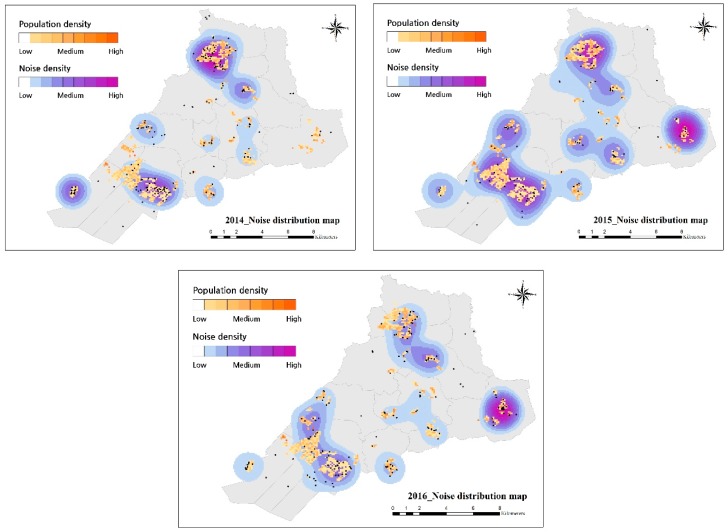
Annually the noise distribution map.

**Figure 9 ijerph-16-01018-f009:**
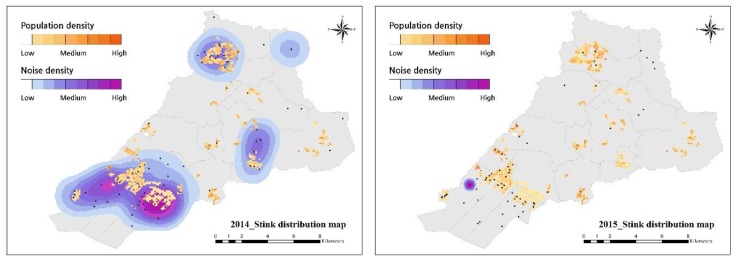
Annually the stink distribution map.

**Table 1 ijerph-16-01018-t001:** Data composition based on reporting channel in Shiheung, Korea.

Year	Daytime Phone Calls	Night-Time Phone Calls	Internet Bulletin Signboard	Smartphone Apps	Sum
2014	78	175	121	55	429
2015	31	111	175	121	438
2016	110	202	176	98	586
sum	219	488	472	274	1453

**Table 2 ijerph-16-01018-t002:** The frequency and the centrality value of degree and betweenness of each word in 2014.

No	Words	Frequency	Degree Centrality	Betweenness Centrality
1	noise	344	0.6875	0.588889
2	stink	241	0.375	0.45
3	construction	141	0.3125	0.048611
4	dawn	106	0.3125	0.251389
5	night-time	95	0.25	0.177778
6	holiday	91	0.1875	-
7	dust	69	0.125	-
8	music	53	0.125	-
9	site	38	0.125	-
10	sleep	34	0.0625	
11	factory	31	0.0625	-
12	window	31	0.0625	-
13	daily life	29	0.0625	-
14	livestock	24	0.0625	-
15	residents	24	0.0625	-
16	pain	20	0.0625	-
17	health	20	0.0625	-

**Table 3 ijerph-16-01018-t003:** The frequency and the centrality value of degree and betweenness of each word in 2015.

No	Words	Frequency	Degree Centrality	Betweenness Centrality
1	stink	451	0.636364	0.81746
2	noise	241	0.181818	0.047619
3	dawn	152	0.181818	0.313853
4	dust	115	0.090909	-
5	construction	103	0.136364	0.041126
6	add facility	78	0.136364	-
7	contaminant	77	0.363636	0.306277
8	nighttime	71	0.045455	-
9	sleep	60	0.090909	-
10	incineration	52	0.181818	0.006854
11	factory	52	0.045455	-
12	window	50	0.045455	-
13	fuel	48	0.181818	0.006854
14	weather	48	0.045455	-
15	pollution	38	0.045455	-
16	gas	38	0.136364	0.031385
17	chemicals	34	0.045455	-
18	parks	32	0.045455	-
19	air	31	0.045455	-
20	pain	30	0.090909	-
21	environment	30	0.045455	-
22	waste water	26	0.045455	-
23	maleficence	21	0.045455	-

**Table 4 ijerph-16-01018-t004:** The frequency and the centrality value of degree and betweenness of each word in 2016.

No	Words	Frequency	Degree Centrality	Betweenness Centrality
1	noise	728	0.809524	0.749546
2	stink	594	0.380952	0.20873
3	construction	255	0.238095	0.102041
4	nighttime	173	0.238095	0.028231
5	dawn	129	0.285714	0.08424
6	dust	126	0.095238	-
7	factory	120	0.095238	0.023469
8	sleep	115	0.142857	-
9	residents	114	0.142857	0.023469
10	holiday	83	0.142857	-
11	inconvenience	72	0.047619	-
12	window	68	0.142857	0.023469
13	residence	68	0.047619	-
14	daily life	51	0.047619	-
15	chemicals	42	0.047619	-
16	door	37	0.047619	-
17	gas	36	0.047619	-
18	music	30	0.047619	-
19	church	27	0.047619	-
20	pain	21	0.047619	-
21	metal	19	0.047619	-

**Table 5 ijerph-16-01018-t005:** The frequency of all the words except for the words that have appeared for three years.

No	Keywords	2014	2015	2016	No	Keywords	2014	2015	2016
1	residents	24	0	114	15	incineration	0	52	0
2	holiday	69	0	83	16	fuel	0	48	0
3	inconvenience	0	0	72	17	weather	0	48	0
4	residence	0	0	68	18	pollution	0	38	0
5	daily life	31	0	51	19	gas	0	38	0
6	chemicals	0	34	42	20	chemicals	0	34	0
7	site	31	0	42	21	parks	0	32	0
8	door	0	0	37	22	air	0	31	0
9	gas	0	38	36	23	environment	0	30	0
10	music	29	0	30	24	waste water	0	26	0
11	church	0	0	27	25	maleficence	0	21	0
12	metal	0	0	19	26	livestock	24	0	0
13	add facility	0	78	0	27	health	20	0	0
14	contaminant	0	77	0	-	-	-	-	-

**Table 6 ijerph-16-01018-t006:** Monthly frequency of the noise complaints.

Month													
**Year**	**Jan**	**Feb**	**Mar**	**Apr**	**May**	**Jun**	**Jul**	**Aug**	**Sep**	**Oct**	**Nov**	**Dec**	**Total**
2014	15	13	22	18	33	51	54	41	42	28	12	15	344
2015	7	15	11	14	17	35	41	37	31	12	11	10	241
2016	42	43	46	58	48	78	72	87	101	56	49	48	728
Total	64	71	79	90	98	164	167	165	174	96	72	73	1313

**Table 7 ijerph-16-01018-t007:** Monthly frequency of stink complaints.

Month													
**Year**	**Jan**	**Feb**	**Mar**	**Apr**	**May**	**Jun**	**Jul**	**Aug**	**Sep**	**Oct**	**Nov**	**Dec**	**Total**
2014	19	10	17	14	13	27	98	25	19	23	22	14	214
2015	27	129	19	27	31	27	25	63	34	34	18	14	451
2016	34	32	38	58	44	54	63	96	70	35	41	29	594
Total	80	171	74	99	88	108	126	184	123	92	81	60	1286

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
