# Peer review of "Analysis of Environmental Issues with an Application of Civil Complaints: The Case of Shiheung City, Republic of Korea"

_ijerph, 2019, doi:10.3390/ijerph16061018_

Reviewer 1 Report

The title of the article is "Analysis of Environmental Issues with Civil complaints; Case of the Shiheung City, Republic of Korea", but enviromental issues in Shisheung City are just briefly presented in paragraph 2.1: a better introduction to what is supposed to be the main issue of the article is needed, also supported by adequate urban maps (location of industrial districts and residential areas etc.).

In the introduction, authors mention the importance of citizens participation for urban policies, but the way citizen participation is intended in this article (civil complaints) is quite questionable. I reccomend to investigate better this topic (by reading more literature, at first S.Airnstein article 'A Ladder Of Citizen Participation',) or at least to better explain the role of citizen participation in urban policies in South Korea because it may be intended with diverse aims and scopes.

Another critical point of this research is the amount of data used to support the thesis: Shiheung City has a population of 500.000 inhabitants ca, but the data collected refer to few hundreds calls/messages. Also the data range (3 years) appears not really useful to understant the link between citizens' complaints and urban/environmental policies, as fig. 7 shows: the noise distribution is mostly the same during the years; it would be more useful to have maps that show (if it happened) how noise distribution change or disappear thanks to citizens' complaints and to ad hoc policies/services. This is a crucial issue, because in the article authors affirm that "opinions reporting by citizens, including civil complaints, is being used as valuable resources  for promoting city services" (line 547-8) but they never show an example of how these complaints are effectively translated into city polices or new services.

Author Response

 Response to Reviewer 1 Comments

Point 1: Environmental issues in Shiheung City are just briefly presented in paragraph 2.1: a better introduction to what is supposed to be the main issue of the article is needed, also supported by adequate urban maps (location of industrial districts and residential areas etc.).

Response 1: As you suggested, we added the issues of the study area in 2.1 part and Figure 2 explained the location of settlement areas and industrial complex. (150~156 line) and Figure 2

Point 2: In the introduction, the authors mention the importance of citizens participation for urban policies, but the way citizen participation is intended in this article (civil complaints) is quite questionable. I recommend at least to better explain the role of citizen participation in urban policies in South Korea because it may be intended with diverse aims and scopes.they may use the proportion approach which will make the interpretation easier.

Response 2: As you suggested, we explained that the civil complaints analysis was reflected in policy in the Republic of Korea, and also described the utilization of analyzed civil complaints. (101~108 line)

Point 3: Another critical point of this research is the amount of data used to support the thesis: Shiheung City has a population of 500.000 inhabitants ca, but the data collected refer to few hundreds calls/messages. Also the data range (3 years) appears not really useful to understant the link between citizens' complaints and urban/environmental policies, as fig. 7 shows: the noise distribution is mostly the same during the years; it would be more useful to have maps that show (if it happened) how noise distribution change or disappear thanks to citizens' complaints and to ad hoc policies/services. This is a crucial issue, because in the article authors affirm that "opinions reporting by citizens, including civil complaints, is being used as valuable resources for promoting city services" (line 547-8) but they never show an example of how these complaints are effectively translated into city polices or new services.

Response 3: As the development-restrict areas were lifted from 2014 to 2016, the city has changed. (169 line). The raw data was 2,328 but the geographical information did not match, or unnecessary contents were deleted during the refining process. (173~174 line). We believe that the analysis of civil complaints is applicable as part of the policy and describe the use of the policy of civil complaints in the Republic of Korea. (101~108 line) Although the data cannot support the onions of all citizens, we think this process will be easy to understand the issues. (624~627 line) The relevant contents in limitations paragraph are described. (636~638 line)

Reviewer 2 Report

The study area should emerge in the abstract. For example, the very general contents in lines 94-103 could be very briefly summarized first in the abstract and then explained in the introduction (as it was done).

The Figure 1 does not well describe the study area. I suggest to improve the caption by giving more precise references in the picture or to put also a satellite map. 

lines 143-144 : it would be useful talking a bit about urban planning tools with the aim of better motivating your sentences about the absence of tools taking into consideration the environmental issues. 

I would advice to insert some pictures of the case study area, especially the places with high density of population (with the word "noise" used very often) in order to allow the reader understand the space the study is referring to.

Is it possible to better identify or summarize who replied to interviews, tests, phone calls, etc.?

Finally, it would be suitable to address some guidelines to policy makers in the par. 4.3.

Author Response

Response to Reviewer 2 Comments

Point 1: The study area should emerge in the abstract. For example, the very general contents in lines 94-103 could be very briefly summarized first in the abstract and then explained in the introduction.

Response 1: As you suggested, we have added and entirely revised the abstract. (13~27 line)

Point 2: The Figure 1 does not well describe the study area. I suggest to improve the caption by giving more precise references in the picture or to put also a satellite map.

Response 2: Satellite map and precise mark was input in Figure 1. And Figure 2 will be explained detailed in study area. (Figure 1 and 2)

Point 3: it would be useful talking a bit about urban planning tools with the aim of better motivating your sentences about the absence of tools taking into consideration the environmental issues (lines 143-144).

Response 3: As you suggested, we described environment consideration in study area. (150~156 line)

Point 4: I would advice to insert some pictures of the case study area, especially the places with high density of population (with the word "noise" used very often) in order to allow the reader understand the space the study is referring to.

Response 4: As you suggested, high density of population is explained to easier understand by Figure 2. (Figure 2)

Point 5: Is it possible to better identify or summarize who replied to interviews, tests, phone calls, etc.?

Response 5: Unfortunately, there is a limit to the data associated with interviews and tests. And data regarding respondents is not include as personal information. However, we would like to add their contents to the next study.

Point 6: It would be suitable to address some guidelines to policy makers in the par. 4.3.

Response 6: As you suggested, the role of policy makers was described in 4.3. (526~528 line)

Reviewer 3 Report

General opinion:

The paper describes a very interesting case of application of information provided voluntarily by citizens. The case study is the Shiheung City (Republic of Korea) and years 2014-2016. The research concerns civil complaints concerning environmental issues.

The collected data are the words extracted from complaints made by citizens using various communication channels. They  show that the most disturbing nuisances/threats/hazards are stink and noise. From the words analysis we can reason that they occur mainly at night or dawn. The authors present both, the noise and stink distribution maps for particular years connecting them with activities taking place in the areas where these nuisances occur.

The research is very interesting, concerning data methods and results. In my opinion three things are missing. First, which ways of communications authors consider as helpful and which of them of little significance. The second issue is how these experiences may be used in other cities or countries. Finally, if it is any further research recommended by the authors.

Detailed remarks:

I suggest considering some title modification, for example “Analysis of Environmental Issues with Application of Civil  Complaints; Case of the Shiheung City, Republic of Korea” or “Application of Civil  Complaints to the Analysis of Environmental Issues; Case of the Shiheung City, Republic of Korea”.

The text is generally easy to understand, but quality of English should be thoroughly improved.

The chapter titles are misspelled:

2.2 – should be “data” instead of “date”

3.2  - should be “Relationship” instead of “Relationshop” 

Meaning of chapter 4.2 title “Reporting channels and continously local issues” is difficult to understand.

The rest of the chapters, as well as figures and tables titles should be reviewed linguistically and corrected or improved.

There is a lot of misspellings. For example:

Line 378 – “decreased” instead of “deceased”

Line 558 – “noise” instead of “nose”.

Some sentences are wrong or difficult to understand. For example …

Lines 562, 563 - “And finally, not all citizens who filed complaint in the city are not all citizens.”

In many fragments, the text is either not correct grammatically or the style is awkward.

The reference [33] is doubled in the line 364.

Figure 1 is not clear. I suggest preparing it in colours.

Author Response

Response to Reviewer 3 Comments

Point 1: First, which ways of communications authors consider as helpful and which of them of little significance.

Response 1: Described the convenience and usefulness of online channels (489-493 line)

Point 2: The second issue is how these experiences may be used in other cities or countries.

Response 2: Described applicability of the process in other fields (546-580 line)

Point 3: Finally, if it is any further research recommended by the authors

Response 3: We propose two ways of mapping in future research; the one is to application of happiness score (‘Headnometer) to words, the other is to compare difference between this process and PPGIS process. (639~646 line)

Point 4: suggest considering some title modification

Response 4: Amended to “Analysis of Environmental Issues with an Application of Civil complaints; Case of Shiheung City, Republic of Korea. (2~4 line)

Point 5: 2.2 – should be “data” instead of “date”?

Response 5: The word was amended to “data” (168 line)

Point 6: 3.2 - should be “Relationship” instead of “Relationshop”?

Response 6: The word was amended to “Relationship” (279 line)

Point 7: Meaning of chapter 4.2 title “Reporting channels and continously local issues” is difficult to understand.

Response 7: The chapter 4.2 title was amended to “Local issues by reporting channels”

(472 line)

Point 8: Line 378 – “decreased” instead of “deceased”?

Response 8: The word was amended to “decreased” (386 line)

Point 9: Line 558 – “noise” instead of “nose”.

Response 9: The word was amended to “noise” (629 line)

Point 10: 3.2 - Lines 562, 563 - “And finally, not all citizens who filed complaint in the city are not all citizens.”

Response 10: The sentence was amended to “These complaints are not representative of all the issues that occur within the city, but civil complaints might be considered typical samples that can be used to improve the urban services process or planning for citizens. (636~638 line)

Point 11: 3.2 - should be “Relationship” instead of “Relationshop”?

Response 11: The doubled reference symbol was removed. (380 line)

Point 12: 3.2 - Figure 1 is not clear. I suggest preparing it in colours..

Response 12: Figure 1 was amended to satellite image and coloring. (148 line, Figure 1)

Reviewer 4 Report

Introduction to this paper is extensive and provides sufficient background for the conducted research. The most important phenomena and processes connected with recent development of the cities and citizens participation are introduced in proper way, however, it might be beneficiary to consider in literature review classics who published work on public participation such as, among others, P. Healey: Collaborative planning: Shaping places in fragmented societies or Gene RoweLynn J. Frewer: Public Participation Methods: A Framework for Evaluation. It might be also useful to end introduction with a very short description what different parts of paper consist of.

Research design steps are adequate and a proper framework is introduced, the content of individual chapters is clearly structured. I would suggest only one minor change - to move section 4.1 from discussion into results, as it includes spatial distribution analyses and not discussion.

Other studies in this area of research is mentioned in a limited scope (e.g. crime in American cites). As there are mostly theoretical principles mentioned in the paper, it might be also useful to analyze if other similar studies of citizens complains were conducted in other parts of the world. In discussion please compare your results with the one from previous studies. The strong part of this study is analysis of strengths and weaknesses of the approach and methods used within this research.

The conclusions drawn on reporting channels and spatial distribution of complains are relevant. I also highly assess the discussion on the role of government and policy recommendation.

Specific comments:

·    line 43 - could you explain term „sustainability flow”?

· line 49 ‘…accompanied exactly analysis, which contributes to positivepersuasion of citizens.’ – did you mean to persuade citizen or to involve them?

·    section 2.1 Study area – the description is very wide and long it might be better to focus on environmental issues in the city, especially on their spatial distribution; it might be also useful to describe previously conducted studies – if such exist - on citizens satisfaction from environmental issues using different methods such as surveys etc. Are the results comparable?

Author Response

Response to Reviewer 4 Comments

Point 1: It might be also useful to end introduction with a very short description what different parts of paper consist of. (Gene Rowe, Lynn J. Frewer: Public Participation Methods: A Framework for Evaluation)

Response 1: Referring to the book editors recommend, we described as citizens’ participation in policy. (62~66 row)

Point 2: I suggest only one minor change - to move section 4.1 from discussion into results, as it includes spatial distribution analyses and not discussion..

Response 2: As we respect your comment, but we'd like to comprehensively explain the monthly frequency, distribution and civil complaints' contents. (4.1 section)

Point 3: It might be also useful to analyze if other similar studies of citizens’ complaints were conducted in other parts of the world. In discussion please compare your results with the one from previous studies

Response 3: As you suggested, we compared and contrasted other paper of Vancouver in Canada case similar to our research flow. (546~580 line)

Point 4: line 43 – “could you explain term „sustainability flow”?

Response 4: Amended to ‘sustainable flow’, that word mean continuously linking between human and environment. (44 line)

Point 5: line 49 ‘…accompanied exactly analysis, which contributes to positive persuasion of citizens.’ – did you mean to persuade citizen or to involve them?

Response 5: This sentence means that urban service and planning should be understandable to citizens (49~50 line).

Point 6: 3.2 - section 2.1 Study area – the description is very wide and long it might be better to focus on environmental issues in the city, especially on their spatial distribution; it might be also useful to describe previously conducted studies – if such exist - on citizens satisfaction from environmental issues using different methods such as surveys etc. Are the results comparable??

Response 6: Unfortunately, there is no study of citizens’ satisfaction and dissatisfaction with environmental issues. However, it added the contents of the evaluated the ecosystem services (137~138 line) and environmental issues within study area (150~156 line).

Round  2

Reviewer 1 Report

The authors have improved the article, providing a better description of the area analysed and clearifying the role of citizens' complaints in the development of public policies in Republic of Korea. They also clarified how these data are exploited by the government.